# Postnatal Expression Profile of MicroRNAs Associated with Cardiovascular Diseases in 3- to 11-Year-Old Preterm-Born Children

**DOI:** 10.3390/biomedicines9070727

**Published:** 2021-06-24

**Authors:** Ilona Hromadnikova, Katerina Kotlabova, Ladislav Krofta, Jan Sirc

**Affiliations:** 1Department of Molecular Biology and Cell Pathology, Third Faculty of Medicine, Charles University, 100 00 Prague, Czech Republic; katerina.kotlabova@lf3.cuni.cz; 2Third Faculty of Medicine, Institute for the Care of the Mother and Child, Charles University, 147 00 Prague, Czech Republic; ladislav.krofta@upmd.eu (L.K.); jan.sirc@upmd.eu (J.S.)

**Keywords:** birth weight of newborns, cardiovascular risk, condition of newborns at the moment of birth, expression, gestational age at delivery, children, microRNA, preterm prelabor rupture of membranes, spontaneous preterm birth, whole peripheral blood

## Abstract

(1) Background: Preterm-born children have an increased cardiovascular risk with the first clinical manifestation during childhood and/or adolescence. (2) Methods: The occurrence of overweight/obesity, prehypertension/hypertension, valve problems or heart defects, and postnatal microRNA expression profiles were examined in preterm-born children at the age of 3 to 11 years descending from preterm prelabor rupture of membranes (PPROM) and spontaneous preterm birth (PTB) pregnancies. The whole peripheral blood gene expression of 29 selected microRNAs associated with cardiovascular diseases was the subject of our interest. (3) Results: Nearly one-third of preterm-born children (32.43%) had valve problems and/or heart defects. The occurrence of systolic and diastolic prehypertension/hypertension was also inconsiderable in a group of preterm-born children (27.03% and 18.92%). The vast majority of children descending from either PPROM (85.45%) or PTB pregnancies (85.71%) had also significantly altered microRNA expression profiles at 90.0% specificity. (4) Conclusions: Postnatal microRNA expression profiles were significantly influenced by antenatal and early postnatal factors (gestational age at delivery, birth weight of newborns, and condition of newborns at the moment of birth). These findings may contribute to the explanation of increased cardiovascular risk in preterm-born children. These findings strongly support the belief that preterm-born children should be dispensarized for a long time to have access to specialized medical care.

## 1. Introduction

Preterm-born children have an increased cardiovascular risk, with the first clinical manifestation during childhood and/or adolescence. Usually, clinical symptomatology appears during early adulthood at the very latest.

In general, multiple risk factors predisposing to a later development of cardiovascular diseases have been identified in preterm-born individuals. These risk factors involve increased peripheral and central systolic (SBP) and/or diastolic (DBP) blood pressures [1,2,3,4,5,6,7,8,9,10,11,12,13,14,15,16,17,18,19,20,21], higher heart rate (HR) [4,22,23], higher fat mass [21], lower functional skin capillary density [4], lower peripheral skin blood flow [24], abnormal retinal vascularization (both structure and function) [1,3,17], increased sympathoadrenal activity together with higher levels of urine catecholamines [22], kidney hypoplasia, incomplete nephrogenesis (reduced number of nephrons) and impaired renal function (decreased glomerular filtration rate, microalbuminuria) [20,25,26,27,28], worsened respiratory parameters usually as a consequence of bronchopulmonary dysplasia (BPD) [19,29,30], impaired exercise capacity [19,30], elevated fasting glucose and cholesterol levels [10], higher serum levels of insulin 2 h after the glucose load [31], decreased insulin sensitivity [32,33,34,35,36], or even higher incidence of systolic or diastolic prehypertension/hypertension [13,25,37,38,39,40], chronic kidney disease [13,25], lipid disorders [39,41], type 1 diabetes mellitus [39,42,43,44,45,46], metabolic syndrome [38], and asthma [29,47,48,49].

Moreover, children, adolescents, or young preterm-born adults were reported to have alterations in both left heart and right heart structure, geometry, and function and alterations in arterial structure and distensibility. Smaller left ventricles [9,12,50], smaller right atria [51], smaller right ventricles [51,52], greater right and left ventricular mass [50,52], smaller ascending aorta diameter [9,18], and higher relative wall thickness [51] were observed in children, adolescents, and young adults born preterm. In addition, narrower abdominal aortas and lower abdominal aortic stiffness were detected together with higher brachial and aortic blood pressures [24]. Besides, lower left ventricular longitudinal shortening and systolic tissue velocity [12], higher transversal shortening fraction [12], lower atrial emptying velocities [12], higher right ventricular myocardial performance index (RVmpi’) [51], elevated aortic wave reflection [14], decreased carotid and brachial distensibility [18], and higher estimated pulmonary vascular resistance (PVR) [51] were found in children, adolescents, and young preterm-born adults.

A strong association between preterm birth before 32 weeks of gestation and the risk of incident heart failure [53] and cerebrovascular disease [54] has been recently reported in children and/or young adults. In parallel, low gestational age at birth has been observed to be associated with increased risk of venous thromboembolism in young adulthood [55]. In addition, an association between low birth weight and the risk of ischemic heart disease and myocardial infarction in young adulthood has been demonstrated, but independently of gestational age at delivery [56].

Besides, children and young adults born extremely preterm had significantly altered neuropsychological outcomes (lower general cognitive abilities, visuomotor skills, prospective memory, and aspects of executive functions and language) [32,57,58]. Gestational age of birth, including preterm birth, has also been observed to be associated with the risk of autism spectrum disorders [59].

Recently, it has been shown that children descending from pregnancies affected with hypertensive disorders (gestational hypertension (GH) or preeclampsia (PE)), metabolic disorders (gestational diabetes mellitus (GDM)), and/or fetal growth restriction (FGR) have altered postnatal expression profiles of microRNAs, playing a role in the pathogenesis of diabetes mellitus and cardiovascular diseases. Altered expression of these microRNAs present in peripheral blood leukocytes of children at the age of 3–11 years may contribute, besides other factors, to the development of diabetes mellitus and cardiovascular diseases later in life [60,61,62].

This study focused on the determination of clinical outcome (the incidence of overweight/obesity, systolic and diastolic prehypertension/hypertension, valve problems and heart defects) in preterm-born children at the age of 3–11 years. The group of preterm-born children consisted of children descending from pregnancies complicated with preterm prelabor rupture of membranes (PPROM) and spontaneous preterm birth (PTB). Preterm-born children descending from pregnancies complicated with other pregnancy-related complications (GH, PE, FGR, GDM, placenta previa, placental abruption, and vaginal bleeding) were not intentionally included in this study. Furthermore, the postnatal microRNA gene expression profile in the whole peripheral blood of preterm-born children was studied. The gene expression of 29 selected microRNAs (miR-1-3p, miR-16-5p, miR-17-5p, miR-20a-5p, miR-20b-5p, miR-21-5p, miR-23a-3p, miR-24-3p, miR-26a-5p, miR-29a-3p, miR-92a-3p, miR-100-5p, miR-103a-3p, miR-125b-5p, miR-126-3p, miR-130b-3p, miR-133a-3p, miR-143-3p, miR-145-5p, miR-146-5p, miR-155-5p, miR-181a-5p, miR-195-5p, miR-199a-5p, miR-210-3p, miR-221-3p, miR-342-3p, miR-499a-5p, and miR-574-3p) involved in the pathogenesis of cardiovascular diseases was the subject of interest [60,61,62,63,64,65].

In addition, the microRNA expression profiles of groups of preterm-born children (descending from either PPROM or PTB pregnancies) and term-born children with regard to the presence of abnormal clinical findings (overweight/obesity, prehypertension/hypertension, valve problems and heart defects) were compared. In parallel, the impact of multiple antenatal and early postnatal factors on postnatal microRNA expression profiles of children was studied. These factors involved therapies applied to mothers to postpone the preterm delivery with the aim to accelerate fetal lung maturation (corticosteroid therapy, antibiotic therapy, and tocolytic therapy), gestational age at delivery, mode of delivery, and clinical parameters of newborns (the birth weight, and the condition of infants at the moment of birth determined using the Apgar scores and pH of cord arterial blood).

To the best of our knowledge, there are currently no data on postnatal expression profiles of microRNAs associated with cardiovascular diseases in preterm-born children if other pregnancy-related complications (GH, PE, FGR, and GDM) are not present. In addition, no data on the long-term impact of corticosteroid, tocolytic, and antibiotic therapies administered to pregnant women at risk of preterm delivery on postnatal microRNA expression profile of preterm-born children are available.

## 2. Materials and Methods

### 2.1. Participants

A total of 355 preterm-born children fulfilling the inclusion criteria of the study were identified in the medical databases of the Institute for the Care of the Mother and Child and were invited to participate in our study. Finally, 111 preterm-born children arrived at examination, which indicates a 31.27% success rate.

The inclusion criteria involved preterm-born children at the age of 3–11 years descending from singleton pregnancies affected with PPROM and PTB. PTB was defined as preterm delivery prior to the 37th gestational week with the occurrence of regular uterine contractions (at least 2 contractions within 10 min), together with cervical changes. In case of PPROM, leakage of amniotic fluid appeared at a minimum of 2 h before the onset of labor, once again prior to the 37th gestational week [66,67,68]. Term-born children were defined as healthy infants weighing more than 2500 g, who were born after 37 completed weeks of gestation to women with a normal course of gestation (no presence of medical, obstetrical, or surgical complications). Term-born children were selected randomly from medical databases on the basis of equal age.

The exclusion criteria involved children descending from preterm birth pregnancies complicated, in addition, with hypertensive disorders, gestational diabetes mellitus, and fetal growth restriction. Concurrently, children affected with inborn defects and chromosomal aneuploidy disorders were not involved in the study. In addition, preterm-born children descending from pregnancies affected with placenta previa, placental abruption, and significant vaginal bleeding were also not included in the study.

The prospective study finally included various ethnic populations of Caucasian children living in the Czech Republic descending from pregnancies with a normal course of gestation (NP, *n* = 92), and pregnancies complicated with PPROM (*n* = 55) and PTB (*n* = 56). The national composition of the inhabitants of the Czech Republic is the following: the Bohemians, the Moravians, the Silesians, the Poles, the Slovaks, the Germans, the Ukrainians, the Hungarians, and the Russians, all belonging to Caucasian population. In the Czech Republic ethnic minorities are represented by the Vietnamese people (Mongoloid race, 0.3%) and Romani people (Gypsies, various ethnic subgroups, 0.1%) (demographic data collected in 2011 during the census). Only a few preterm-born children from these ethnic minority groups were delivered in the Institute for the Care of the Mother and Child, Prague, and none of them were interested in participating in our study. The clinical characteristics of preterm- and term-born children are summarized in Table 1.

Maternal prenatal corticosteroid administration to accelerate fetal lung maturation was applied in 82 out of 111 women at risk of preterm delivery [69,70,71]. Prophylactic antibiotic therapy was administered to 86 out of 111 women at risk of preterm delivery [72,73]. The tocolytic therapy was applied, only if non-contraindicated, to 64 out of 111 women at risk of preterm delivery [72,74,75].

The case group involved 19 extremely preterm children (delivered before the 28th gestational week), 33 very preterm children (delivered within the 28th and 32nd gestational weeks) and 59 moderate to late preterm children (delivered within the 32nd and 37th gestational weeks) [72,75,76]. Forty-six preterm-born children were delivered via cesarean section and 65 children were delivered via vaginal birth.

The Apgar scores were determined after the birth of newborns at 1 min, 5 min, and 10 min [77,78,79,80]. A reassuring Apgar score after five minutes (7–10) was detected in 101 preterm-born infants. A moderately abnormal Apgar score after five minutes (4–6) was detected in six preterm-born infants, and a concerning Apgar score after five minutes (0–3) was detected in three preterm-born infants. Significant metabolic acidosis (umbilical cord arterial blood pH < 7.0) was detected in 7 out of 111 preterm-born infants.

Informed consents were signed by legal representatives of the children included in the study. The Ethics Committees (the Institute for the Care of the Mother and Child and the Third Faculty of Medicine, Charles University, Prague, Czech Republic) approved the study (grant no. AZV 16-27761A, “Long-term monitoring of complex cardiovascular profile in the mother, fetus and offspring descending from pregnancy-related complications”, dates of approval: 27 March 2014 and 28 May 2015). All procedures were in compliance with the Helsinki declaration of 1975, as revised in 2000.

### 2.2. Clinical Examination of Children

Standardized BMI assessment, BP measurement and an ultrasound of the heart (echocardiography) were performed as described previously [60,61,62]. At least three BP measurements were performed on different occasions. Children with normal BP had average SBP and DBP values below the 90th percentile for age, sex, and height. Children with prehypertension had repeatedly average systolic or diastolic BP within the range of ≥90th percentile and <95th percentile. Children with hypertension had during three independent measurements average systolic or diastolic BP ≥ 95th percentile [81].

The BMI in children was assessed using the age-specific and sex-specific BMI percentile calculator [82]. Children with normal BMI values ranged within >5th percentile and <85th percentile. Overweight children had BMI within the range of >85th percentile and <95th percentile. Obese children were those with BMI ≥95th percentile.

A complete two-dimensional echocardiography was performed by a pediatrician experienced in echocardiography using a Philips HD15 ultrasound machine (Philips Ultrasound, Bothell, WA, USA) and a sector array transducer (3–8 MHz) incorporating color flow, pulse-wave, and continuous-wave Doppler measurements with adaptive technology. Preterm- and term-born children with abnormal clinical findings were referred to the Department of Pediatric Cardiology.

### 2.3. Processing of Samples and Relative MicroRNA Quantification via Real-Time RT-PCR

Processing of samples, reverse transcription, and relative quantification of microRNAs were done as previously described [60,61,62,63,64,65]. In brief, homogenized cell lysates were prepared from samples of unclotted whole peripheral venous blood (200 µL) as soon as possible after blood collection using a QIAamp RNA Blood Mini kit (Qiagen, Hilden, Germany, no. 52304).

Total RNA was isolated using a mirVana microRNA Isolation kit (Ambion, Austin, TX, USA, no. AM1560) and treated with DNase I (Thermo Fisher Scientific, Carlsbad, CA, USA, no. EN0521).

Reverse transcription was performed using microRNA-specific stem–loop RT primers, the components of TaqMan MicroRNA Assays, and a TaqMan MicroRNA Reverse Transcription kit (Applied Biosystems, Branchburg, NJ, USA, no. 4366597) in a total reaction volume of 10 µL on a 7500 Real-Time PCR system (Applied Biosystems, Branchburg, NJ, USA) under the following conditions: 30 min at 16 °C, 30 min at 42 °C, 5 min at 85 °C, and then held at 4 °C.

Relative quantification of microRNAs was performed using real-time PCR in a total reaction volume of 15 µL, in which case cDNA (3 µL) was mixed with specific TaqMan MGB primers and probes (TaqMan MicroRNA Assay; Applied Biosystems, Branchburg, NJ, USA), and the constituents of TaqMan Universal PCR Master Mix (Applied Biosystems, Branchburg, NJ, USA, no. 4318157). The samples were tested in duplicate and regarded as positive if the amplification occurred at Ct <40 (Ct, threshold cycle). The expression of the studied microRNAs was determined using the comparative Ct method [83], and normalized to endogenous controls with the lowest expression variability between the studied samples (geometric mean of RNU58A and RNU38B) [84]. A reference sample, the fetal part of one randomly selected placenta of normally ongoing pregnancy, was used throughout the study for relative microRNA quantification.

### 2.4. Processing of Experimental Data

Logistic regression was used to compare the presence of abnormal clinical findings (the incidence of overweight/obesity, systolic and diastolic prehypertension/hypertension, valve problems or heart defects) among the groups of preterm and term-born children. Since microRNA gene expression data did not show a normal distribution using the Shapiro–Wilk test [85], microRNA levels were compared among the appropriate groups using non-parametric tests (Mann–Whitney test and Kruskal–Wallis test).

Using receiver operating characteristic (ROC) curves, the areas under the curves (AUC), the optimal cut-off points, and the respective sensitivities at 10.0% false positive rate (FPR) for the studied microRNAs were assessed (MedCalc Software bvba, Ostend, Belgium). The *p*-values of *p* < 0.05 were considered statistically significant. Logistic regression combined with ROC curve analysis was used to assess the optimal combinations of microRNA biomarkers (MedCalc Software bvba, Ostend, Belgium). In brief, in this setting, the power of the model’s predicted values to discriminate between positive and negative cases is quantified by the area under the ROC curve. To perform a full ROC curve analysis, the predicted probabilities are first saved and then used as a new variable in ROC curve analysis. The dependent variable used in logistic regression then acts as the classification variable in the ROC curve analysis dialog box.

The box plots representing log-normalized expression values [quantitative reverse transcription polymerase chain reaction (RT-qPCR) expression, log_10_ 2^−ΔΔCt^] for selected individual microRNAs with the best sensitivity at 10.0% FPR are presented.

Correlation between variables was calculated using the Spearman rank correlation coefficient (ρ). If the correlation coefficient values ranged within <−1.0; −0.5>, there was a strong negative correlation. If it was within the interval <−0.5; 0>, there was a weak negative correlation.

### 2.5. Information on MicroRNA–Gene–Biological Pathways/Disease Interactions

The MiRWalk database and the Predicted Target module were used to provide information on predicted targets of microRNAs dysregulated in the whole peripheral blood of preterm-born children [86].

MiRWalk is a comprehensive database that provides information on microRNAs from humans, mice, and rats on their predicted and/or validated target genes. miRWalk2.0 not only documents miRNA binding sites within the complete sequence of a gene, but also combines this information with a comparison of binding sites from 12 existing miRNA-target prediction programs (DIANA-microTv4.0, DIANA-microT-CDS, miRanda, mirBridge, miRDB4.0, miRmap, miRNAMap, DoRiNA, PicTar2, PITA, RNA22v2, RNAhybrid2.1, and Targetscan6.2) to build novel, comparative platforms of binding sites for the promoter (4 prediction datasets), cds (5 prediction datasets), 5′-UTR (5 prediction datasets), and 3′-UTR (13 prediction datasets) regions. Information on the miRNA-target interactions of 2035 disease ontologies (DO), 6727 human phenotype ontologies (HPO), and 4980 OMIM disorders is available. This information provides possible interactions between the microRNAs and genes associated with the 597 KEGG, 456 Panther, and 522 Wiki pathways.

## 3. Results

### 3.1. Clinical Outcomes in Preterm- and Term-Born Children

The groups of children with abnormal clinical findings consisted of those already dispensarized in the Department of Pediatric Cardiology; those indicated, by the sonographer during a visit to our department, to have valve problems and/or heart defects; those confirmed, over several visits to our department, to have either systolic prehypertension/hypertension and/or diastolic prehypertension/hypertension; and/or those who were diagnosed to be overweight/obese (NP, *n* = 39/92; preterm birth, *n* = 66/111). The group of children with abnormal echocardiogram findings consisted of those indicated to have the following valve problems and/or heart defects: tricuspid valve regurgitation (NP, *n* = 8/89; preterm birth, *n* = 14/111), mitral valve regurgitation (NP, *n* = 1/89; preterm birth, *n* = 0/111), pulmonary valve regurgitation (NP, *n* = 2/89; preterm birth, *n* = 12/111), bicuspid aortic valve regurgitation (NP, *n* = 1/89; preterm birth, *n* = 0/111), ventricular septum defect (NP, *n* = 1/89; preterm birth, *n* = 0/111), atrial septum defect (NP, *n* = 1/89; preterm birth, *n* = 0/111), foramen ovale apertum (NP, *n* = 5/89; preterm birth, *n* = 11/111), arrhythmia (NP, *n* = 1/89; preterm birth, *n* = 0/111), and ductus arteriosus patens (NP, *n* = 0/89; preterm birth, *n* = 3/111).

Normal clinical findings (normal anamnesis, normal BP, normal BMI, and normal reference values of echocardiographic measurements) were found in 45 out 111 preterm-born children and in 53 out of 92 term-born children.

Logistic regression revealed a higher incidence of valve problems and/or heart defects and a higher incidence of systolic prehypertension/hypertension in a group of preterm-born children (PPROM and PTB). There was also a trend toward a higher incidence of diastolic prehypertension/hypertension in a group of preterm-born children (PPROM and PTB). However, no difference in the incidence of overweight/obesity between the groups of preterm and term-born children was observed (Table 2).

### 3.2. Postnatal MicroRNA Expression Profiles in Preterm-Born Children (PPROM and PTB)

Firstly, postnatal microRNA gene expression profiles in preterm-born children (children descending from pregnancies complicated with preterm prelabor rupture of membranes and spontaneous preterm birth) were assessed. The overwhelming majority of the tested microRNAs (miR-1-3p, miR-16-5p, miR-17-5p, miR-20a-5p, miR-20b-5p, miR-21-5p, miR-23a-3p, miR-24-3p, miR-26a-5p, miR-29a-3p, miR-100-5p, miR-103a-3p, miR-125b-5p, miR-126-3p, miR-130b-3p, miR-133a-3p, miR-143-3p, miR-145-5p, miR-146-5p, miR-181a-5p, miR-195-5p, miR-199a-5p, miR-221-3p, miR-499a-5p, and miR-574-3p) showed significantly increased expression in children descending from either PPROM or PTB pregnancies (Appendix A).

MicroRNA gene expression profiles differentiated between children descending from PPROM pregnancies and term-born children with a sensitivity (range 16.36–78.18%) at 10.0% FPR (Appendix A). Likewise, microRNAs were able to differentiate between children descending from PTB pregnancies and term-born children with a sensitivity (range 17.86–73.21%) at 10.0% FPR (Appendix A).

Screening based on a combination of 10 selected microRNAs with the best sensitivity (miR-1-3p, miR-20a-5p, miR-20b-5p, miR-26a-5p, miR-100-5p, miR-103a-3p, miR-126-3p, miR-143-3p, miR-195-5p, and miR-499a-5p) (Appendix A) was able to identify at 10.0% FPR 85.45% children descending from PPROM pregnancies with a potential cardiovascular risk (AUC 0.951, *p* < 0.001, sensitivity 85.45%, specificity 91.30%, cut-off >0.3310) (Figure 1).

In parallel, screening based on the combination of 13 selected microRNAs with the best sensitivity (miR-1-3p, miR-16-5p, miR-17-5p, miR-20a-5p, miR-20b-5p, miR-26a-5p, miR-29a-3p, miR-103a-3p, miR-126-3p, miR-143-3p, miR-181a-5p, miR-195-5p, and miR-499a-5p) (Appendix A) showed the best accuracy for children descending from PTB pregnancies. This combined microRNA screening revealed a potential cardiovascular risk in 85.71% children descending from PTB pregnancies at 10.0% FPR (AUC 0.958, *p* < 0.001, sensitivity 89.29%, specificity 86.96%, cut-off >0.2255) (Figure 2).

### 3.3. Postnatal MicroRNA Expression Profiles in Preterm-Born Children (PPROM and PTB) with Regard to the Absence or Presence of Abnormal Clinical Findings (Overweight/Obesity, Prehypertension/Hypertension, Valve Problems and Heart Defects)

Furthermore, microRNA expression profiles between groups of preterm-born children (descending from either PPROM or PTB pregnancies) and term-born children with regard to the presence and the absence of abnormal clinical findings (overweight/obesity, prehypertension/hypertension, valve problems and heart defects) were compared.

Irrespective of the absence or the presence of abnormal clinical findings, an increased expression of miR-1-3p, miR-16-5p, miR-17-5p, miR-20a-5p, miR-20b-5p, miR-21-5p, miR-26a-5p, miR-29a-3p, miR-100-5p, miR-103a-3p, miR-125b-5p, miR-126-3p, miR-130b-3p, miR-133a-3p, miR-143-3p, miR-146-5p, miR-181a-5p, miR-195-5p, miR-199a-5p, miR-221-3p, miR-499a-5p, and miR-574-3p was observed in both groups of preterm-born children when compared to term-born children [PPROM, normal or abnormal clinical findings (Appendix A, Appendix A); PTB, normal or abnormal clinical findings (Appendix A, Appendix A)].

In addition, an increased expression of miR-23a-3p and miR-145-5p was observed in both groups of preterm-born children with the presence of abnormal clinical findings (PPROM (Appendix A); PTB (Appendix A)) and in PTB born children with the presence of normal clinical findings (Appendix A) when compared to term-born children. Moreover, upregulation of miR-24-3p was present only in preterm-born children with abnormal clinical findings (PPROM (Appendix A); PTB (Appendix A)).

### 3.4. Postnatal MicroRNA Expression Profiles in Preterm- and Term-Born Children with Regard to Gestational Age at Delivery

Each group of preterm-born children subdivided into individual subcategories based on the gestational age at delivery (extremely preterm, very preterm, and moderate to late preterm) showed significantly increased postnatal microRNA gene expression (miR-1-3p, miR-16-5p, miR-17-5p, miR-20a-5p, miR-20b-5p, miR-21-5p, miR-26a-5p, miR-29a-3p, miR-100-5p, miR-103a-3p, miR-125b-5p, miR-126-3p, miR-130b-3p, miR-133a-3p, miR-143-3p, miR-146a-5p, miR-181a-5p, miR-195-5p, miR-199a-5p, miR-221-3p, miR-499a-5p, and miR-574-3p) when compared with term-born children (Appendix A, Appendix A). On the other hand, increased postnatal levels of miR-23a-3p (extremely preterm, and moderate to late preterm), miR-24-3p (moderate to late preterm), and miR-145-5p (very preterm and moderate to late preterm) reached statistical significance only in some groups of preterm-born children. Nevertheless, there was no difference in microRNA gene expression between the groups of extremely preterm-, very preterm-, and moderate to late preterm-born children.

Moreover, in a total group of preterm and term-born children, strong or weak negative correlations between postnatal microRNA gene expression (miR-1-3p, miR-16-5p, miR-17-5p, miR-20a-5p, miR-20b-5p, miR-21-5p, miR-23a-3p, miR-24-3p, miR-26a-5p, miR-29a-3p, miR-100-5p, miR-103a-3p, miR-125b-5p, miR-126-3p, miR-130b-3p, miR-133a-3p, miR-143-3p, miR-145-5p, miR-146-5p, miR-181a-5p, miR-195-5p, miR-199a-5p, miR-221-3p, miR-499a-5p, and miR-574-3p) and the gestational age at delivery were observed (Table 3). This finding indicates that preterm-born children have significantly higher postnatal levels of microRNAs than term-born children.

### 3.5. Postnatal MicroRNA Expression Profiles in Preterm- and Term-Born Children with Regard to Birth Weight of Newborns

In parallel, in a total group of preterm and term-born children, strong or weak negative correlations between postnatal microRNA gene expression (miR-1-3p, miR-16-5p, miR-17-5p, miR-20a-5p, miR-20b-5p, miR-21-5p, miR-23a-3p, miR-24-3p, miR-26a-5p, miR-29a-3p, miR-100-5p, miR-103a-3p, miR-125b-5p, miR-126-3p, miR-130b-3p, miR-133a-3p, miR-143-3p, miR-145-5p, miR-146-5p, miR-181a-5p, miR-195-5p, miR-199a-5p, miR-221-3p, miR-499a-5p, and miR-574-3p) and the birth weight of newborns were observed (Table 4). This finding indicates that preterm-born children with lower birth weight have significantly higher postnatal levels of microRNAs than full term children with normal birth weight.

### 3.6. Postnatal MicroRNA Expression Profiles in Preterm-Born Children (PPROM and PTB) with Regard to Mode of Delivery

No association between the mode of delivery (vaginal birth and cesarean birth) and postnatal microRNA gene expression was observed in preterm-born children (PPROM and PTB).

### 3.7. Postnatal MicroRNA Expression Profiles in Preterm-Born Children (PPROM and PTB) with Regard to Condition of Newborns at Moment of Birth

No association between the Apgar scores (AS) determined at 5 min and 10 min after the birth of newborns and postnatal microRNA gene expression was observed in preterm-born children with the exception of miR-133a-3p, which showed a weak negative correlation with the Apgar scores (AS) determined at 5 min (ρ = −0.227; *p* = 0.017) and 10 min (ρ = −0.192; *p* = 0.045) after the birth of newborns.

In addition, a weak negative correlation between the pH of cord arterial blood, a parameter of metabolic acidosis in newborns, and postnatal gene expression of miR-133a-3p (ρ= −0.222; *p*= 0.019), miR-199a-5p (ρ= −0.254; *p*= 0.007), and miR-221-3p (ρ= −0.189; *p*= 0.047) was observed in preterm-born children.

### 3.8. Postnatal MicroRNA Expression Profiles in Preterm-Born Children (PPROM and PTB) with Regard to Antenatal Application of Corticosteroids, Antibiotics, and Tocolytics to Mothers

The antenatal administration of corticosteroids to mothers with the aim to accelerate lung maturation in the fetus had no impact on the postnatal microRNA gene expression profile in preterm-born children (PPROM and PTB). In parallel, the treatment with tocolytics to postpone premature labor also had no impact on the postnatal microRNA gene expression profile in preterm-born children (PPROM and PTB). Similarly, no association between the antibiotic administration to women in preterm delivery and postnatal microRNA gene expression was observed in preterm-born children (PPROM and PTB).

### 3.9. Information on MicroRNA–Gene–Biological Pathways/Disease Interactions

Information on microRNA–gene–biological pathways/disease interactions was provided on microRNAs dysregulated in the whole peripheral blood of preterm-born children. Predicted targets of microRNAs involved in key human biological pathways, with a role in the pathogenesis of preterm delivery, were reported. These biological pathways involve apoptosis, inflammatory response, senescence, and autophagy. In addition, predicted targets of microRNAs associated with cardiovascular risk factors (hypertension, congenital heart defects, patent foramen ovale, patent ductus arteriosus, and heart valve disease) that appeared more frequently in our group of preterm-born children were also reported.

A large group of genes (predicted targets) of cardiovascular disease-associated microRNAs aberrantly expressed in the whole peripheral blood of preterm-born children is also involved in key biological processes related to pathogenesis of preterm delivery, such as the apoptosis pathway, an inflammatory response pathway, senescence, and autophagy pathways (Table 5 and Table 6).

In addition, predicted targets of microRNAs aberrantly expressed in the whole peripheral blood of preterm-born children indicated that a large group of genes is also involved in the pathogenesis of diseases, which increase cardiovascular risk (hypertension, congenital heart defects, patent foramen ovale, patent ductus arteriosus, and heart valve disease) (Table 7 and Table 8).

## 4. Discussion

Initially, a comprehensive physical examination of preterm- and term-born children delivered in our hospital to obtain baseline medical information was performed. Standardized BMI assessment, BP measurement, and an ultrasound of the heart (echocardiography) were performed in children at the age of 3 to 11 years. From the clinical examination, it was evident that nearly one-third of preterm-born children descending from PPROM or PTB pregnancies (32.43%) had valve problems and/or heart defects that should be checked at least by pediatric cardiologists and if needed the children should be dispensarized to have access to specialized medical care.

In addition, the occurrence of systolic prehypertension/hypertension (27.03%) and diastolic prehypertension/hypertension (18.92%) was also inconsiderable in a group of preterm-born children (PPROM or PTB) when compared with term-born children. No child had a confirmed diagnosis of chronic hypertension or was on blood pressure treatment, when physical examination was performed. Definitely, children indicated to have prehypertension/hypertension during several visits should be checked from a long-term perspective to exclude an accidental BP increase resulting from the visit to the medical department or to confirm the diagnosis of chronic hypertension.

A certain concern arises also from the fact that the incidence of overweight/obesity was nearly the same within the groups of age-matched preterm and term-born children in spite of the substantially lower weight of preterm-born newborns.

Nevertheless, the findings resulting from this study were consistent with the findings of other researchers who identified increased peripheral SBP, increased peripheral DBP and/or a trend toward increased peripheral DBP in preterm-born children [4,6,7,8,10,11,12]. The data were also in compliance with the observation of another scientific group [40], which found no differences in BMI and waist–hip ratio between full term, healthy preterm and sick preterm young adults. Nevertheless, this study was not held during childhood, but at early adulthood, at the age of 23 years [40].

With the exception of a study reporting a higher prevalence of congenital heart disease inclusive of atrial and ventricular septal defects and patent ductus arteriosus in preterm-born school children [87] and a study demonstrating a higher prevalence of interarterial septal aneurysm in young adults born preterm [88], no data on the incidence of valve problems (tricuspid, mitral, pulmonary, and bicuspid aortic valve regurgitations) or other heart defects (foramen ovale apertum, and arrhythmia) are available in preterm-born children older than 3 years.

Furthermore, postnatal microRNA expression profiles were compared between preterm and term-born children. Preterm-born children descending from pregnancies with other parallel complications were not involved in the study by intent, since postnatal microRNA gene expression in children prenatally affected with fetal growth restriction, maternal hypertensive and metabolic disorders had already been investigated separately [60,61,62]. Concurrently, preterm-born children, whose mothers had labor complicated with placenta previa, placental abruption, and significant vaginal bleeding, were also not included in the study.

The majority of the tested microRNAs (25 out of 29 microRNAs) showed higher expression levels in the whole peripheral blood of preterm-born children, irrespective of the condition which led to preterm delivery (PPROM or PTB). The postnatal screening based on only one microRNA, miR-1-3p, reached a very high accuracy to discriminate between preterm and term-born children. In total, 78.18% children descending from PPROM pregnancies and 73.21% children descending from PTB pregnancies showed increased postnatal miR-1-3p expression at 10.0% FPR (Appendix A, Appendix A).

Finally, 10 microRNAs with the best accuracy to differentiate between PPROM and term-born children were selected for a postnatal screening (miR-1-3p, miR-20a-5p, miR-20b-5p, miR-26a-5p, miR-100-5p, miR-103a-3p, miR-126-3p, miR-143-3p, miR-195-5p, and miR-499a-5p). In the case of PTB born children, 13 microRNAs with the best differentiation accuracy were at last selected for a postnatal screening (miR-1-3p, miR-16-5p, miR-17-5p, miR-20a-5p, miR-20b-5p, miR-26a-5p, miR-29a-3p, miR-103a-3p, miR-126-3p, miR-143-3p, miR-181a-5p, miR-195-5p, and miR-499a-5p). Overall, at 10.0% FPR, a combined screening identified the vast majority of preterm-born children descending from either PPROM (85.45%) or PTB (85.71%) pregnancies to have altered postnatal microRNA expression profiles.

So far, this finding is comparable with the finding in children descending from GDM pregnancies, where our group identified, through combination screening, altered postnatal microRNA expression profiles in 88.14% of the children [61]. With respect to this fact, preterm-born children descending from either PPROM or PTB pregnancies have, next to children descending from GDM pregnancies, the most serious alterations in postnatal microRNA expression profiles out of all groups of children affected with pregnancy-related complications [60,61,62].

On the grounds of these clinical and experimental findings, we strongly support the efforts of other investigators suggesting the long-term dispensarization of preterm-born children as a highly risky group for a later development of cardiovascular diseases deserving regular monitoring, implementation of preventive measures and therapeutic interventions if needed.

In addition, we studied the relation between postnatal microRNA gene expression and multiple antenatal and early postnatal factors, including gestational age at delivery, birth weight of newborns, the condition of newborns at the moment of birth, mode of delivery, and antenatal application of corticosteroids, antibiotics, and tocolytics to mothers. It was apparent that both gestational age at delivery and birth weight of newborns were significantly associated with postnatal levels of microRNAs in preterm- and term-born children. The highest postnatal microRNA expression levels were observed in children born prematurely at lower gestational ages and in children born with lower birth weight. These findings may also contribute to the explanation of why preterm-born children with lower birth weight have an increased cardiovascular risk and to the explanation of the increase in cardiovascular risk with the decrease in gestational age at delivery [2,5,9,10,11,15,19,20,22,23,26,27,29,31,35,39,43,44,51,58,59].

Similarly, the condition of newborns at the moment of birth also had an impact on the postnatal microRNA expression profiles of preterm-born children, even though this impact was not as significant as the impact of gestational age at delivery and birth weight of newborns. miR-133a-3p showed a weak negative correlation with the Apgar scores determined at 5 min and 10 min and the pH of cord arterial blood after the birth of newborns. In addition, a weak negative correlation between the pH of cord arterial blood and the postnatal gene expression of miR-199a-5p and miR-221-3p was observed in preterm-born children. These findings indicate that preterm-born children with the worst values of parameters assessed immediately after the delivery may have higher postnatal levels of some cardiovascular disease-associated microRNAs. These findings may also contribute to the explanation of why preterm-born children have an increased cardiovascular risk.

On the other hand, the administration of tocolytic drugs, the administration of antenatal corticosteroids, and the administration of antibiotics to mothers at the time of preterm delivery had no impact on the postnatal microRNA expression profiles of preterm-born children. Likewise, the mode of delivery (vaginal or cesarean birth) had no influence on the postnatal microRNA expression profiles of preterm-born children.

On the grounds of these data, we suppose that altered microRNA expression profiles in the whole peripheral blood of preterm-born children might be associated with increased cardiovascular risk. We believe that the dysregulation of cardiovascular disease-associated microRNAs in preterm-born children might be induced by in utero exposure to a gestation with a pathological course. The pathogenesis of preterm delivery is multifactorial, but some principal pathogenic mechanisms such as cervical insufficiency, uterine malformations, acute inflammation of the membranes and chorion of the placenta followed by an exaggeration of inflammatory processes have already been discovered [68,89,90,91,92,93,94]. Moreover, premature aging of the fetal membranes encompassing telomere shortening, senescence, apoptosis, and proteolysis has also been demonstrated to play a key role in the pathogenesis of preterm delivery [95,96,97]. Interactions between cardiovascular disease-associated microRNAs with altered expression profiles in the whole peripheral blood of preterm-born children and specific genes involved in key biological processes related to pathogenesis of preterm delivery, such as the apoptosis pathway, inflammatory response pathway, senescence, and autophagy pathways, were demonstrated in this study.

In addition, some diseases identified to have a higher prevalence in our cohort of preterm-born children might also contribute to alterations of microRNA expression profiles and might even increase the present risk of later development of cardiovascular diseases. Recently, the occurrence of valve problems and heart defects has been demonstrated to intensify the upregulation of most microRNAs (miR-1-3p, miR-16-5p, miR-17-5p, miR-20a-5p, mir-20b-5p, miR-21-5p, miR-26a-5p, miR-29a-3p, miR-100-5p, miR-125b-5p, miR-126-3p, miR-143-3p, miR-146a-5p, miR-181a-5p, miR-195-5p, miR-221-3p, miR-499a-5p, and miR-574-3p) already present in the whole peripheral blood of children exposed to complicated pregnancies [62].

Although the association between the presence of prehypertension/hypertension (combined systolic and diastolic prehypertension/hypertension and/or isolated systolic or diastolic prehypertension/hypertension) and the expression of cardiovascular disease-associated microRNAs has not yet been identified in the association analysis previously performed in children born from normal and complicated pregnancies [62], we assume that the presence of chronic hypertension might exacerbate current alterations and induce de-novo alterations in microRNA expression profiles in preterm-born children similarly as in young and middle-aged mothers with confirmed diagnosis of systolic hypertension and on blood pressure treatment [65]. Previously, it was demonstrated that 7 out of 29 tested microRNAs (miR-17-5p, miR-24-3p, miR-92a-3p, miR-126-3p, miR-130b-3p, miR-181a-5p, and miR-210-3p) showed increased expression in mothers with systolic hypertension when the comparison to mothers with normal SBP values was performed regardless of the course of previous gestation (women with a history of normal and complicated pregnancies). The expression of microRNAs (miR-24-3p, miR-210-3p, miR-221-3p, and miR-342-3p) also differed between mothers on blood pressure treatment and without medication [65]. In parallel, an association between the presence of hypertension and altered expression of cardiovascular disease-associated microRNAs (miR-21-5p, miR-100-5p, miR-103a-3p, miR-145-5p, miR-181a-5p, and miR-199a-5p) was observed by other investigators [98,99,100,101,102,103,104]. However, large cohort studies involving a sufficient number of preterm-born children with the confirmed diagnosis of chronic prehypertension/hypertension are needed to prove the association between the presence of chronic prehypertension/hypertension and altered microRNA expression profiles.

Interactions between microRNAs dysregulated in the whole peripheral blood of preterm-born children and specific genes related to human disease ontologies, such as hypertension, congenital heart defects, patent foramen ovale, patent ductus arteriosus, and heart valve disease, were also demonstrated in this study. Nevertheless, further studies are definitely needed to evaluate the possible role of postnatal microRNA expression as a predictor for a later development of cardiovascular diseases in preterm-born children.

This study has some limitations. A set of 29 microRNAs, known to be involved in the pathogenesis of cardiovascular diseases, was selected for our previous and currently ongoing studies [60,61,62,63,64,65]. However, the list of selected microRNAs did not include all the microRNAs known to be involved in the pathogenesis of cardiovascular diseases. In addition, the list of selected microRNAs for our previous and currently ongoing studies [60,61,62,63,64,65] was not based on genome-wide expression studies since these analyses had not been performed in the cohorts of our patients. In addition, another possible limitation of this study is the fact that, while dysregulated microRNAs play a role in the functioning of the cardiovascular system, there are certainly other risk factors that can contribute to the development of cardiovascular diseases.

## 5. Conclusions

In conclusion, nearly one-third of preterm-born children descending from either PPROM or PTB pregnancies had valve problems and/or heart defects. The occurrence of prehypertension/hypertension was also inconsiderable in a group of preterm-born children. A vast majority of preterm-born children also had severe postnatal alterations in microRNA expression profiles at the age of 3 to 11 years. These postnatal microRNA expression profiles were significantly influenced by many antenatal and early postnatal factors. Among these factors are conditions leading to preterm birth, gestational age at delivery, birth weight of newborns, and the condition of newborns at the moment of birth. These findings may contribute to the explanation of the increased cardiovascular risk in preterm-born children. These findings strongly support the belief that preterm-born children should be dispensarized for a long time to have access to specialized medical care.

## 6. Patents

National patent granted—Industrial Property Office, Czech Republic (Patent No. 308102).

International patent filed—Industrial Property Office, Czech Republic (PCT/CZ2019/050050).

## Figures and Tables

**Figure 1 biomedicines-09-00727-f001:**
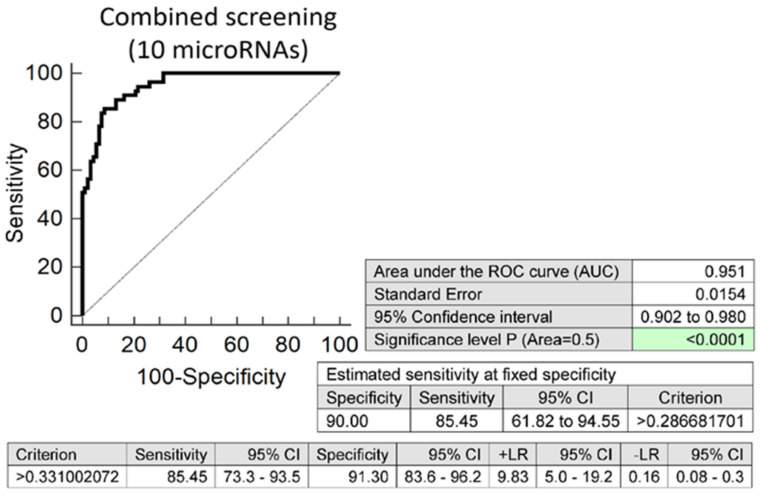
Altered postnatal microRNA expression profile in children descending from PPROM pregnancies. Combined screening of 10 selected microRNAs with the best sensitivity (miR-1-3p, miR-20a-5p, miR-20b-5p, miR-26a-5p, miR-100-5p, miR-103a-3p, miR-126-3p, miR-143-3p, miR-195-5p, and miR-499a-5p) reveals 85.45% children with a potential cardiovascular risk at 10.0% FPR.

**Figure 2 biomedicines-09-00727-f002:**
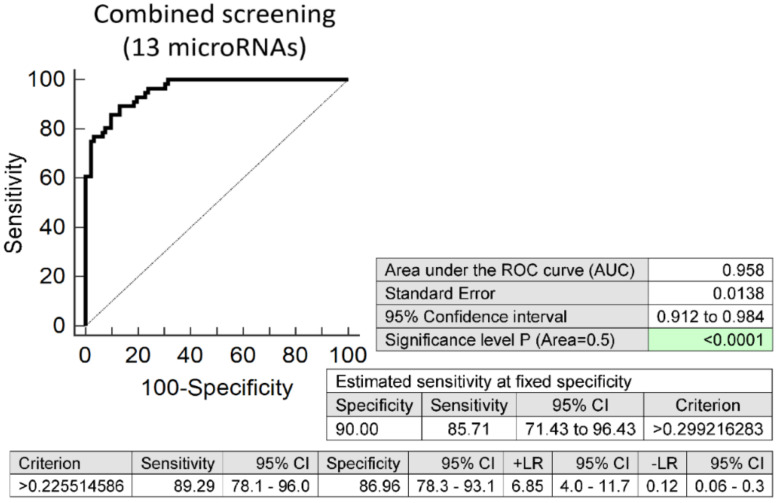
Altered postnatal microRNA expression profile in children descending from PTB pregnancies. Combined screening of 13 selected microRNAs with the best sensitivity (miR-1-3p, miR-16-5p, miR-17-5p, miR-20a-5p, miR-20b-5p, miR-26a-5p, miR-29a-3p, miR-103a-3p, miR-126-3p, miR-143-3p, miR-181a-5p, miR-195-5p, and miR-499a-5p) reveals 85.71% children with a potential cardiovascular risk at 10.0% FPR.

**Table 1 biomedicines-09-00727-t001:** Characteristics of cases and controls.

	Normal Pregnancies (*n* = 92)	PPROM (*n* = 55)	PTB (*n* = 56)	*p*-Value ^1^	*p*-Value ^2^
**Children at follow-up**					
Age (years)	5.74 ± 1.90	6.25 ± 1.71	6.30 ± 1.78	0.151	0.118
Height (cm)	118.96 ± 12.99	121.92 ± 11.90	121.71 ± 10.30	0.344	0.315
Weight (kg)	22.75 ± 6.05	24.02 ± 6.34	23.39 ± 6.23	0.532	1.000
BMI (kg/m^2^)	15.73 ± 1.53	15.87 ± 2.01	15.50 ± 2.15	1.000	0.680
Systolic BP (mmHg)	100.98 ± 8.60	105.47 ± 8.47	103.68 ± 7.79	0.009	0.213
Diastolic BP (mmHg)	61.13 ± 7.33	63.95 ± 6.76	62.82 ± 6.22	0.052	0.433
Heart rate (*n*/min)	89.63 ± 13.59	91.89 ± 10.73	92.09 ± 11.91	1.000	1.000
Abnormal clinical findings	39 (42.39%)	66 (59.46%)		**0.015**	
**During gestation**					
Maternal age at delivery (years)	32.50 ± 4.13	32.20 ± 4.81	31.48 ± 4.36	1.000	1.000
GA at delivery (weeks)	39.91 ± 0.96	32.38 ± 2.50	30.45 ± 3.67	**<0.001**	**<0.001**
Mode of delivery					
Vaginal	81 (88.04%)	26 (47.27%)	39 (69.64%)	**<0.001**	**0.005**
CS	11 (11.96%)	29 (52.73%)	17 (30.36%)		
Fetal birth weight (g)	3398.42 ± 366.72	1908.51 ± 518.96	1610.41 ± 668.38	**<0.001**	**<0.001**
Fetal sex					
Boy	47 (51.09%)	22 (40.0%)	37 (66.07%)	0.192	0.074
Girl	45 (48.91%)	33 (60.0%)	19 (33.93%)		
Primiparity					
Yes	49 (46.74%)	37 (67.27%)	33 (58.93%)	0.095	0.501
No	43 (53.26%)	18 (32.73%)	23 (41.07%)		
Birth order of index pregnancy					
1st	39 (42.39%)	26 (47.27%)	21 (37.50%)	0.543	0.201
2nd	34 (36.96%)	16 (29.09%)	19 (33.93%)		
3rd	15 (16.30%)	8 (14.55%)	8 (14.29%)		
4th+	4 (4.35%)	5 (9.09%)	8 (14.29%)		
Infertility treatment					
Yes	3 (3.26%)	9 (16.36%)	9 (16.07%)	**0.005**	**0.006**
No	89 (96.74%)	46 (83.64%)	47 (83.93%)		
Antenatal application of corticoids to mothers					
Yes	-	43 (78.18%)	39 (69.64%)	-	-
No	-	12 (21.82%)	17 (30.36%)	-	-
Antenatal application of antibiotics to mothers					
Yes	-	51 (92.73%)	35 (62.50%)	-	-
No	-	4 (7.27%)	21 (37.50%)	-	-
Antenatal application of tocolytics to mothers					
Yes	-	28 (50.91%)	36 (64.29%)	-	-
No	-	27 (49.09%)	20 (35.71%)	-	-
Apgar score < 7; 5 min	0	5 (9.09%)	4 (7.14%)	-	-
Apgar score < 7; 10 min	0	4 (7.27%)	1 (1.79%)	-	-
Umbilical blood pH	7.30 ± 0.10	7.29 ± 0.10	7.27 ± 0.13	0.673	1.000

Data are presented as mean ± SD (standard deviation) for continuous variables and as number (percent) for categorical variables. Statistically significant results are marked in bold. Continuous variables are compared using the Kruskal–Wallis test. Categorical variables are compared using a chi-square test. *p*-value ^1^: the comparison of children descending from normal pregnancies and children descending from PPROM pregnancies; *p*-value ^2^: the comparison of children descending from normal pregnancies and children descending from PTB pregnancies. PPROM, preterm prelabor rupture of membranes; PTB, spontaneous preterm birth; BP, blood pressure; CS, cesarean section; GA, gestational age.

**Table 2 biomedicines-09-00727-t002:** Clinical outcomes of preterm- and term-born children.

	Normal Pregnancy	Preterm Birth Pregnancy	OR (95% CI)	*p*-Value
Overweight/obesity	10/92 (10.87%)	11/111 (9.91%)	0.902 (0.365–2.229)	0.823
Systolic prehypertension/hypertension	11/91 (12.09%)	30/111 (27.03%)	2.694 (1.264–5.741)	0.007
Diastolic prehypertension/hypertension	9/91 (9.89%)	21/111 (18.92%)	2.126 (0.921–4.906)	0.068
Valve problems or heart defects	17/89 (19.10%)	36/111 (32.43%)	2.089 (1.077–4.051)	0.026

Logistic regression is used to compare the incidence of abnormal clinical findings between the groups of preterm- and term-born children. The significance level is established at a *p*-value of < 0.05. A higher incidence of valve problems and/or heart defects is observed in a group of preterm-born children (PPROM and PTB). In addition, a higher incidence of systolic prehypertension/hypertension is observed in a group of preterm-born children (PPROM and PTB). Diastolic prehypertension/hypertension shows a trend toward statistical significance in a group of preterm-born children (PPROM and PTB). No difference in the incidence of overweight/obesity is observed between the groups of preterm and term-born children.

**Table 3 biomedicines-09-00727-t003:** Postnatal increase in microRNA gene expression with descending gestational age at delivery.

microRNA	ρ	*p*-Value
miR-1-3p	−0.680874	*p* < 0.001
miR-16-5p	−0.417812	*p* < 0.001
miR-17-5p	−0.368344	*p* < 0.001
miR-20a-5p	−0.494584	*p* < 0.001
miR-20b-5p	−0.415185	*p* < 0.001
miR-21-5p	−0.394461	*p* < 0.001
miR-23a-3p	−0.235916	*p* < 0.001
miR-24-3p	−0.205205	*p* = 0.003
miR-26a-5p	−0.400153	*p* < 0.001
miR-29a-3p	−0.501671	*p* < 0.001
miR-92a-3p	−0.030566	*p* = 0.665
miR-100-5p	−0.401612	*p* < 0.001
miR-103a-3p	−0.441161	*p* < 0.001
miR-125b-5p	−0.381679	*p* < 0.001
miR-126-3p	−0.498944	*p* < 0.001
miR-130b-3p	−0.337410	*p* < 0.001
miR-133a-3p	−0.427561	*p* < 0.001
miR-143-3p	−0.509350	*p* < 0.001
miR-145-5p	−0.252306	*p* < 0.001
miR-146a-5p	−0.406642	*p* < 0.001
miR-155-5p	0.050472	*p* = 0.313
miR-181a-5p	−0.441077	*p* < 0.001
miR-195-5p	−0.459024	*p* < 0.001
miR-199a-5p	−0.408683	*p* < 0.001
miR-210-3p	0.062054	*p* = 0.379
miR-221-3p	−0.433073	*p* < 0.001
miR-342-3p	−0.066637	*p* = 0.345
miR-499a-5p	−0.496558	*p* < 0.001
miR-574-3p	−0.380311	*p* < 0.001

ρ, Spearman’s rank correlation coefficient.

**Table 4 biomedicines-09-00727-t004:** Postnatal increase in microRNA gene expression with descending birth weight of newborns.

microRNA	ρ	*p*-Value
miR-1-3p	−0.696108	*p* < 0.001
miR-16-5p	−0.443317	*p* < 0.001
miR-17-5p	−0.396386	*p* < 0.001
miR-20a-5p	−0.520593	*p* < 0.001
miR-20b-5p	−0.464224	*p* < 0.001
miR-21-5p	−0.396125	*p* < 0.001
miR-23a-3p	−0.223679	*p* = 0.001
miR-24-3p	−0.198504	*p* = 0.005
miR-26a-5p	−0.415544	*p* < 0.001
miR-29a-3p	−0.528150	*p* < 0.001
miR-92a-3p	−0.070582	*p* = 0.317
miR-100-5p	−0.400367	*p* < 0.001
miR-103a-3p	−0.451351	*p* < 0.001
miR-125b-5p	−0.385954	*p* < 0.001
miR-126-3p	−0.523181	*p* < 0.001
miR-130b-3p	−0.340302	*p* < 0.001
miR-133a-3p	−0.454867	*p* < 0.001
miR-143-3p	−0.529926	*p* < 0.001
miR-145-5p	−0.284582	*p* < 0.001
miR-146a-5p	−0.414050	*p* < 0.001
miR-155-5p	0.027801	*p* = 0.594
miR-181a-5p	−0.462912	*p* < 0.001
miR-195-5p	−0.471084	*p* < 0.001
miR-199a-5p	−0.399153	*p* < 0.001
miR-210-3p	0.013085	*p* = 0.853
miR-221-3p	−0.462300	*p* < 0.001
miR-342-3p	−0.078768	*p* = 0.264
miR-499a-5p	−0.523619	*p* < 0.001
miR-574-3p	−0.406839	*p* < 0.001

ρ, Spearman’s rank correlation coefficient.

**Table 5 biomedicines-09-00727-t005:** A list of predicted targets of microRNAs dysregulated in the whole peripheral blood of preterm-born children in relation to the apoptosis pathway, using the miRWalk2.0 database (data available in the KEGG, Wiki, and Panther pathways).

	Predicted Targets		
microRNA	KEGG Pathways	Wiki Pathways	Panther Pathways
miR-1	IKBKB, IL3, PIK3R5	CASP2, IKBKB, LTA	ATF2, BAG4, BCL2L10, IKBKB, MAP4K2, MAP4K3, PRKCE, PRKCG
miR-16-5p	BCL2, IKBKB, IRAK2, PRKAR1A, PRKAR2A	BCL2, IKBKB	BCL2, CRADD, IKBKB
miR-17-5p	ATM, CASP6, CASP7, CASP8, CASP10, CFLAR, CYCS, DFFA, EXOG, FASLG, IL1R1, IRAK1, IRAK4, MAP3K14, PIK3R2, PPP3CA, PRKAR2A, PRKX, TNFRSF10A, TNFRSF10D, XIAP	BNIP3L, CASP6, CASP7, CASP8, CASP10, CFLAR, CYCS, DFFA, FASLG, IRF1, TNFRSF1B, TNFRSF21, XIAP	ATF6, BAG1, CASP7, CASP8, CASP10, CFLAR, CREB1, CREM, CYCS, EIF2S1, FASLG, HSPA5, MAP3K14, MAPK9, PRKCQ, REL, TNFRSF10D, TNFRSF1B, XIAP
miR-20a-5p	CHP2, PPP3R1	BCL2L11, CASP2, MCL1, TNFRSF21, TP73	BCL2L11, HSPA6, HSPA8, MAP3K5, MCL1, PRKCA, TMBIM6
miR-20b-5p	ATM, CASP6, CASP7, CASP8, CASP10, CFLAR, CYCS, DFFA, EXOG, FASLG, IL1R1, IRAK1, IRAK4, MAP3K14, PIK3R2, PPP3CA, PRKAR2A, PRKX, TNFRSF10A, TNFRSF10D, XIAP	BNIP3L, CASP6, CASP7, CASP8, CASP10, CFLAR, CYCS, DFFA, FASLG, IRF1, TNFRSF1B, TNFRSF21	ATF6, BAG1, CASP7, CASP8, CASP10, CFLAR, CREB1, CREM, CYCS, EIF2S1, FASLG, HSPA5, MAP3K14, MAPK9, PRKCQ, REL, TNFRSF10D, TNFRSF1B, XIAP
miR-21-5p	APAF1, CFLAR, FASLG	APAF1, CFLAR, FASLG, MAP3K1	APAF1, CFLAR, DAXX, EIF2S1, FASLG, MAP2K3
miR-23a-3p	CFLAR, EXOG, IKBKB, PIK3CB, TNFRSF10, TNFSF10B	CFLAR, IGF1, IKBKB, MAP3K1, TNFRSF10, TNFSF10B	BIK, CFLAR, CREM, EIF2AK2, IKBKB, MAP3K5, PIK3CB, TNFRSF10, TNFSF10B
miR-24-3p	BCL2L1, EXOG, FASLG, IKBKB, IL1B, IRAK4, MYD88, PIK3CB, RIPK1	BBC3, BCL2L2, BCL2L11, BNIP3L, FASLG, IKBKB, MYC, NFKBIE, RIPK1, TRAF1, TRAF3	BCL2L1, BCL2L2, BCL2L11, EIF2AK2, FASLG, FOS, IKBKB, PIK3CB, PRKCA, PRKCH, RIPK1
miR-26a-5p	APAF1, BID, BIRC2, CASP6, DFFB, PPP3CB, PPP3CC	APAF1, BAK1, BID, BIRC2, CASP6, CRADD, DFFB, MDM2, PMAIP1	APAF1, ATF2, BAG4, BAK1, BID, BIRC2, CRADD, CREB1, EIF2AK2, FOS, HSPA8, PRKCD, PRKCQ, RELB
miR-29a-3p	CASP8, CYCS, IL1RAP, TNFRSF1A	BAK1, CASP8, CYCS, HRK, IGF1, MCL1, TNFRSF1A	BAK1, CASP8, CYCS, HSPA5, MCL1, TNFRSF1A
miR-100-5p	IRAK3, PPP3CA	-	RELB
miR-103a-3p	IL1RAP, IL3, PRKAR1A	BCL2L2, CASP2, CRADD, IRF1, IRF5, TNFRSF25	ATF6, BCL2L10, BCL2L2, BIK, CRADD, HSPA1B, LTB, MADD, MAP2K3, MAPK3, PRKCD
miR-125b-5p	AIFM1, CAPN1, CASP9, CSF2RB, EXOG, IKBKG	BAK1, CASP2, CASP9, IKBKG, IRF4, MCL1, PRF1	AIFM1, BAG4, BAK1, CASP9, MADD, MCL1, PRKRA, REL, TMBIM6
miR-126-3p	TNFRSF10B	TNFRSF10B	-
miR-130b-3p	CHUK, PIK3CA	CHUK, TNFRSF1B, TP73	CREB1, CHUK, PIK3CA, TNFRSF1B
miR-133a-3p	ENDOD1, IRAK3, MAP3K14, TNFRSF10B	BCL2L2, BNIP3L, TNFRSF10B	BCL2L2, MAP3K14, TNFRSF10B
miR-143-3p	APAF1, BIRC2, BIRC3, TNFRSF10B, TNFRSF10D	APAF1, BIRC2, BIRC3, TNFRSF10B	APAF1, BAG1, BIRC2, BIRC3, MAPK3, MAPK9, PRKCE, TNFRSF10D
miR-145-5p	AIFM1, PIK3R5, TNFRSF10B	TNFRSF10B, TNFRSF25	AIFM1, MAP4K2, TMBIM6, TNFRSF10B
miR-146a-5p	CASP7, CASP9, DFFA, IL3, IRAK1, IRAK4, PPP3R2, PRKACA	CASP2, CASP7, CASP9, DFFA, PMAIP1, PRF1	BAG1, CASP7, CASP9, HSPA1A, JDP2, PRKCE
miR-181a-5p	AKT3, ATM, CASP8, CSF2RB, ENDOD1, EXOG, IL1A, IL1R1, IL1RAP, PPP3R1, PRKAR2A, TRADD	CASP8, CRADD, IRF5, MDM2, PMAIP1, TP63, TRADD	AKT3, ATF2, CASP8, CRADD, DAXX, FOS, MAPK1, TRADD
miR-195-5p	BCL2, IKBKB, IRAK2, PRKAR1A, PRKAR2A	BCL2, CRADD, IKBKB	BCL2, CRADD, IKBKB
miR-199a-5p	IKBKG, PRKAR1A, PRKX, RELA, TNF, TRADD	BBC3, GZMB, IKBKG, RELA, TNF, TRADD, TRAF1	CREM, EIF2AK2, GZMB, MAPK9, PRKCA, RELA, RELB, TNF, TRADD
miR-221-3p	AKT3, APAF1, CASP10, IKBKG, IL1RAP, PIK3CD, PPP3R1, TNFSF10	APAF1, BNIP3L, CASP10, IKBKG, IRF4, MAPK10, MDM2, TNFSF10	AKT3, APAF1, ATF2, ATF4, CASP10, CREB1, MAPK10, PIK3CD, PRKCB, TNFSF10
miR-499a-5p	AKT2, IL1RAP, PIK3CD, PPP3CA, PRKAR1A	MDM2	AKT2, ATF2, HSPA8, PIK3CD, PRKCE, TMBIM6
miR-574-3p	-	TP63	LTB, MADD

**Table 6 biomedicines-09-00727-t006:** A list of predicted targets of microRNAs dysregulated in the whole peripheral blood of preterm-born children in relation to the senescence and autophagy pathways and the inflammatory response pathway, using the miRWalk2.0 database (data available in Wiki pathways).

	Predicted Targets (Wiki Pathways)	
microRNA	Senescence and Autophagy Pathways	Inflammatory Response Pathway
miR-1	ATG13, FN1, IL, LAMP2	CD28, FN1, IL2RB
miR-16-5p	BCL2, CREG1, HMGA1, LAMP2, MAP2K1, RAF1, SMAD4	IL2RA
miR-17-5p	ATG10, ATG12, CD44, CDKN1A, E2F1, IL6R, IRF1, LAMP2, RNASEL, RSL1D1, SERPINE1, SH3GLB1	CD28, IL5, LAMC1, LAMC2, TNFRSF1B
miR-20a-5p	ATG14,ATG16L1, ATG5, ATG7, BECN1, BRAF, IGFBP7, IL8, RNASEL, RSL1D1, SH3GLB1, SQSTM1, ULK1	-
miR-20b-5p	ATG10, ATG12, CD44, CDKN1A, E2F1, IL6R, IRF1, LAMP2, RNASEL, RSL1D1, SERPINE1, SH3GLB1	CD28, IL5, LAMC1, LAMC2, TNFRSF1B
miR-21-5p	MAP2K3	THBS3
miR-23a-3p	AMBRA1, ATG13, BECN1, IFNG, IGF1, IL6R, IL8, MAPK14, PLAU	IFNG
miR-24-3p	ATG13, CDKN1B, FN1, IFNG, IGFBP5, IL1B, IL6R, MAP1LC3A, MAP1LC3C, MMP14	CD28, CD86, FN1, IFNG, IL2RB, LAMC1
miR-26a-5p	ATG13, COL10A1, HMGA1, IFNG, IL6, MDM2, PCNA, PTEN, RB1, ULK1	COL1A2, IFNG
miR-29a-3p	IGF1, PTEN, RNASEL, SH3GLB1	COL1A2, IL2RA, LAMC1, TNFRSF1A
miR-100-5p	MTOR	-
miR-103a-3p	ATG14, GABARAPL1, HMGA1, IL3, IRF1, IRF5, MAP2K3, SERPINB2, SMAD4	CD40
miR-125b-5p	AKT1S1, IGFBP3, RAF1	IL2RB
miR-126-3p	-	-
miR-130b-3p	AMBRA1, ATG14, CD44, IGFBP5, KMT2A, MAP2K1, MLST8, PTEN, RNASEL, SERPINE1	LAMB2, TNFRSF1B
miR-133a-3p	ATG14, FN1, GABARAPL1, MAPK14, MMP14, RB1CC1, SLC39A1	CD28, FN1
miR-143-3p	ATG10, CD44, HMGA1, IFNG, IGFBP5, SERPINE1, SLC39A3	CD28, CD40, IFNG, IL2RA
miR-145-5p	AMBRA1, CD44, HMGA1, IFNB1, MAP1LC3B, SLC39A2	IL2RA
miR-146a-5p	ATG12, IL3, KMT2A, RNASEL, SERPINB2, TNFSF15	CD80, CD86
miR-181a-5p	ATG10, ATG5, CDKN1B, CXCL1, IL1A, IRF5, MAPK1, MDM2, PTEN, ULK1	COL1A2, IL2, IL2RB, LAMC1
miR-195-5p	BCL2, CREG1, HMGA1, LAMP2, MAP2K1, RAF1, SMAD4	IL2RA
miR-199a-5p	CEBPB, IGFBP3, IL6, SLC39A3, UVRAG	IL4R
miR-221-3p	-	THBS1, VTN
miR-499a-5p	ATG3, MDM2, RB1	IL2RB, IL5RA
miR-574-3p	-	IL2RB

**Table 7 biomedicines-09-00727-t007:** A list of predicted targets of microRNAs dysregulated in the whole peripheral blood of preterm-born children in relation to hypertension, using the miRWalk2.0 database (data available in the Disease Search database).

	Predicted Targets
microRNA	Disease Search
miR-1	ACSM3, ARG2, BDNF, CCL2, DRD1, EDN1, KCNA5, KCNMA1, LTA, MEX3C, NR3C1, PTPN1, SLC6A19, SLC8A1, SLCO1B1, TTR
miR-16-5p	ALOX12, APLN, ATP2A2, CTH, CXCL10, GHR, HTR2A, KL, NISCH, P2RY2, SGK1, SLC12A2, VEGFA, XDH
miR-17-5p	ACVRL1, ADORA2B, ADRA1A, APEX1, ATP2A2, ATP2B1, BMP7, BMPR2, CASP8, CD36, CX3CL1, CYP4F2, ECE1, FLT1, GSTM3, HIF1A, HTR2A, IAPP, IL10, IL23R, KCNMA1, KDR, KLC1, KYNU, LEP, MMP2, MTHFR, NOS1, NOX4, OPTN, PPARA, PRCP, PTGIS, RGS2, ROCK2, SELP, SERPINE1, SLC22A3, SLC26A4, SLC2A5, SLC6A4, SLC6A9, SMAD5, SREBF1, STAT3, TNFRSF1B, TRPM6, VEGFA, XDH
miR-20a-5p	ABCA1, ATP1A2, CD36, CYP3A5, DRD1, ERAP1, ESR1, F3, FLT1, FMO3, GUCY1A3, HTR2A, MFN2, MMP3, NOX1, RLN1, SLC12A3, SMAD1, TGFBR2, TRPC4, TTR
miR-20b-5p	ACVRL1, ADORA2B, ADRA1A, APEX1, ATP2A2, ATP2B1, BMP7, BMPR2, CASP8, CD36, CX3CL1, CYP4F2, ECE1, FLT1, GSTM3, HIF1A, HTR2A, IAPP, IL10, IL23R, KCNMA1, KDR, KLC1, KYNU, LEP, MMP2, MTHFR, NOS1, NOX4, OPTN, PPARA, PRCP, PTGIS, RGS2, ROCK2, SELP, SERPINE1, SLC22A3, SLC26A4, SLC2A5, SLC6A4, SLC6A9, SMAD5, SREBF1, STAT3, TNFRSF1B, TRPM6, TRPM7, VEGFA, XDH
miR-21-5p	CXCL10, EDNRB, KLF5, PPARA, THPO
miR-23a-3p	ABCA1, ADRA2B, ARG1, ATP1A1, CXCL12, EDNRB, HSD11B1, IGF1, KLF5, MAPK14, MBL2, PRCP, TRPM6, ZNF652
miR-24-3p	ADD2, ADM, ADRA1A, ANPEP, ATP2A2, BDKRB2, CPS1, CX3CR1, GOSR2, HMOX1, HSD11B2, IL1B, KLC1, MAT1A, NOS3, NPR1, PIM1, PPARG, SLC12A3, SLC7A1, TSHR, WNK1
miR-26a-5p	ADM, ATP1A2, CALCRL, CCL2, CRP, CTH, ENG, ESR1, FLT1, HGF, HLA-A, IER3, IL6, NOS1, PCNA, PIM1, PON1, PTGS2, PTX3, ROBO4, SELP, SLC12A2, SLCO4C1, SMAD1, STK39, TAP1, TRPC3, TRPC4, WNK1, XDH, ZNF652
miR-29a-3p	APLNR, CASP8, CXCL10, IGF1, KNG1, LEP, LPL, NOS2, SDK1, SGK1, VEGFA, XPNPEP1
miR-100-5p	-
miR-103a-3p	ADRA1A, ADRB2, ALAD, ALOX12, APLN, BDNF, CD40, CHGA, CRP, CYP2C9, DRD1, EDNRB, FURIN, HSPA1B, HTR2A, KLC1, LIPE, MFN2, MLYCD, MMP3, NISCH, NOS1, RNLS, ROBO4, SLC22A2, SLC6A2, SLC7A1, SLCO4C1, TGFBR3, UCP2, UMOD, VWF, XDH
miR-125b-5p	ADRA1A, ANPEP, BDKRB2, BMPR1B, CCR2, CCR5, CYP11B2, EMILIN1, ENPEP, ENPP1, EPO, GNB3, KCNK3, LEP, MLYCD, MMP2, PSMB9, ROBO4, STAT3
miR-126-3p	-
miR-130b-3p	ABCA1, ADD2, ARG2, ATP1A2, CAPN10, CSK, F3, HLA-A, HLA-B, IL15, IL23R, MBL2, MEF2A, OTC, SERPINE1, STEAP4, TH, TNFRSF1B
miR-133a-3p	ADRA2B, ALAD, CHGB, CXCL12, GOSR2, KCNMA1, KYNU, MAPK14, PEPD, SCG2, SLC22A2
miR-143-3p	ADRB2, ARHGEF1, CCR2, CD40, CYP11B1, CYP1A2, CYP2C9, ENPP1, EPO, FGB, GHR, GNB3, IER3, IL18, KLF5, NFKBIL1, PDE5A, SERPINC1, SLC12A3, SREBF1, UMOD, VNN1, ZNF652
miR-145-5p	ACSM3, ADRA1A, ESR2, GCGR, INPPL1, MAP1LC3B, SCG2, SELP, THPO, TRPM6
miR-146a-5p	CAT, CFH, CYP11A1, HSPA1A, INPPL1, KCNK3, KCNMA1, NOS1, NOX4, PRCP, PTGS2, PTPN1, RHOA, THPO, WISP1, WNK4, ZNF652
miR-181a-5p	AR, ATP1B1, ATP2B1, AVPR1A, CACNB2, CASP8, CTH, CYP4F2, EDNRA, F11R, F3, GREM1, HSD3B1, IL1A, KCNMA1, KL, LEPR, MAOA, MAPK1, NOS3, NR3C1, PTGS2, SLC7A1, SOD3, STEAP4, TGFBR2, VIP, WISP1, ZNF652
miR-195-5p	ALOX12, APLN, ATP2A2, CTH, CXCL10, GHR, HTR2A, KL, NISCH, P2RY2, SGK1, SLC12A2, VEGFA, XDH
miR-199a-5p	CCR2, CHGA, CSK, DDAH2, F11R, GGT1, HTR2A, IL6, MAT1A, MMP9, PTGIS, SCG2, TNC, TNF, VCAM1
miR-221-3p	ATP1A1, CD36, KLC1, MTR, RNLS, VCAM1
miR-499a-5p	CACNB2, CLU, CYP3A4, FLT1, GUCY1A3, NR3C1, SLC12A2, SLC22A8
miR-574-3p	ID2, LEP, MMP3, NISCH, ZNF652

**Table 8 biomedicines-09-00727-t008:** A list of predicted targets of microRNAs dysregulated in the whole peripheral blood of preterm-born children in relation to congenital heart defects and heart valve disease, using the miRWalk2.0 database (data available in the Disease Search database and the OMIM Disorder database).

		Predicted Targets		
**microRNA**	**OMIM Disorder and Disease Search** (congenital heart defects; nonsyndromic 1, X-linked; nonsyndromic 2)	**Disease Search** (patent ductus arteriosus)	**Disease Search** (patent foramen ovale)	**Disease Search** (heart valve disease)
miR-1	GJA1, JARID2	-	-	-
miR-16-5p	ATP2A2	-	-	-
miR-17-5p	ATP2A2, GJA1, MTHFR	MTHFR	-	-
miR-20a-5p	ATP2A2, F3	-	-	-
miR-20b-5p	ATP2A2, GJA1, MTHFR	MTHFR	-	-
miR-21-5p	JAG1	-	-	-
miR-23a-3p	CBS	-	-	-
miR-24-3p	ATP2A2, STRA6, TAB2	MYH11	-	-
miR-26a-5p	CBS, JAG1	-	-	-
miR-29a-3p	-	-	-	-
miR-100-5p	-	-	-	-
miR-103a-3p	SLN	CYP2C9	-	-
miR-125b-5p	-	-	-	-
miR-126-3p	-	-	-	-
miR-130b-3p	ACVR1, F3, JARID2, STRA6	-	-	-
miR-133a-3p	ZIC3	-	-	VKORC1
miR-143-3p	CRELD1	-	-	-
miR-145-5p	-	-	-	-
miR-146a-5p	JAG1	TFAP2B	-	-
miR-181a-5p	F3, SLN	-	-	-
miR-195-5p	ATP2A2	-	-	-
miR-199a-5p	MTHFD1, STRA6	-	-	-
miR-221-3p	MTR	-	-	-
miR-499a-5p	ZIC3	-	-	-
miR-574-3p	-	-	-	-

## Data Availability

The data presented in this study are available on request from the corresponding author. The data are not publicly available due to rights reserved by funding supporters.

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
