# Peer review of "Postnatal Expression Profile of MicroRNAs Associated with Cardiovascular Diseases in 3- to 11-Year-Old Preterm-Born Children"

_biomedicines, 2021, doi:10.3390/biomedicines9070727_

Round 1

Reviewer 1 Report

Hromadnikova and coauthors investigated the Postnatal expression profile of microRNAs associated with cardiovascular diseases in 3-11 years preterm born children.

The authors concluded that “Nearly one third of preterm born children descending from either PPROM or PTB pregnancies had valve problems and/or heart defects. The occurrence of prehypertension/hypertension was also inconsiderable in a group of preterm born children. A vast majority of preterm born children had also severe postnatal alterations in microRNA expression profile at the age of 3 to 11 years. These postnatal microRNA expression profiles were significantly influenced by many antenatal and early postnatal factors. Among these factors belong at least conditions leading to preterm birth, gestational age at delivery, birth weight of newborns, and condition of newborns at moment of birth. These findings may contribute to the explanation of increased cardiovascular risk in preterm born children. These findings strongly support the beliefs that preterm born children should be long-term dispenzarised to have access to specialized medical care.

This is one excellent study on an important topic. In general, the manuscript well prepared and the study carefully designed and mapped out. This study brings very important knowledge regarding the postnatal expression profile of microRNAs associated with cardiovascular diseases in preterm born children. These results may trace pathways for better screening tools in identifying preterm born children at risk developing cardiovascular diseases.

These findings are interesting and important to the field. But I have some concerns detailed below:

The authors should proofread their manuscript by a professional writer.

The title of the manuscript should be changed to a more precise title like: Postnatal expression profile of microRNAs associated with cardiovascular diseases in 3-11 years preterm born children

There is a redundancy in the Introduction which seems too long and could be reorganized and shortened.

Data is convincing, but not well-presented. Can the authors present data in the tables in a graph format?

Is there any association between preterm born children, microRNAs and psychiatric disorders (anxiety, learning and memory, developmental, eating, etc.) and metabolic syndrome in these school-aged children?

Since cerebral blood flow was not accessed in any given time, authors should remove cerebrovascular disease association from their manuscript.

Is there any correlation between mother’s demography and preterm born children and microRNAs? Mother’s demography should be provided.

Typos errors in lines 515 and 537 should be fixed.

Reviewer 2 Report

Please include a more clear aim on the introduction.

Thank you for the opportunity to review this study. The study has a number of significant limitations that may be possible to correct and amend. This is a generally well-written paper that addresses an important researched area.  

Some recommendations for change, and some minor corrections are described below:

Methods: Add inclusion and exclusion criteria

The process of sample recruitment 

Rewrite table, including Mean ± SD

Study design should be improved

Replace the p-values with p<0.05 or p>0.05 when necessary

Include limitations of the study and more in further studies in the discussion.

Paragraphs should be bigger
